# Influences of Toothbrushing and Different Toothpastes on the Surface Roughness and Color Stability of Interim Prosthodontic Materials

**DOI:** 10.3390/ma15175831

**Published:** 2022-08-24

**Authors:** Ayşegül Köroğlu, Onur Şahin, Ahmet Serkan Küçükekenci, Doğu Ömür Dede, Hüsniye Yıldırım, Burak Yilmaz

**Affiliations:** 1Department of Prosthodontics, Faculty of Dentistry, Zonguldak Bülent Ecevit University, Zonguldak 67600, Turkey; 2Department of Prosthodontics, DCT Clinic, Antalya 07000, Turkey; 3Department of Prosthodontics, Faculty of Dentistry, Ordu University, Ordu 52200, Turkey; 4Department of Prosthodontics, Faculty of Dentistry, Nuh Naci Yazgan University, Kayseri 38100, Turkey; 5Department of Reconstructive Dentistry and Gerodontology, School of Dental Medicine, University of Bern, 3012 Bern, Switzerland; 6Department of Restorative, Preventive and Pediatric Dentistry, School of Dental Medicine, University of Bern, 3012 Bern, Switzerland; 7Division of Restorative and Prosthetic Dentistry, The Ohio State University, Columbus, OH 43210, USA

**Keywords:** interim dental material, surface roughness, toothbrush abrasion, toothpaste, color stability

## Abstract

The surface properties and color stability of interim crown materials may vary depending on the toothbrushing procedure. This study aimed to investigate the effects of toothbrushing and different toothpastes on the surface roughness (R_a_) and color stability of different interim crown materials. Disc-shaped specimens were prepared from four interim crown materials (Tab 2000 (ChPM), Imident (LaPM), Protemp 4 (ChDM), and Telio-CAD (CadPM)). Specimens were divided into four subgroups for the control group (Cnt) and for simulated toothbrushing with distilled water (Dw) or with two different toothpastes (whitening toothpaste (WTp), activated charcoal toothpaste (ACTp)). The specimens’ R_a_ values were measured before and after 10,000 cycles of toothbrushing. The color parameters were measured and the color differences (ΔE_00_) were calculated. Data were statistically analyzed by two-way analysis of variance (ANOVA) and Tukey’s HSD tests. A significant increase in the R_a_ values was observed after toothbrushing, except for the LaPM_Dw, ChDM_Dw, and all the CadPM specimens (*p* < 0.05). Toothbrushing with toothpastes increased the ΔE_00_ values of all ChPM and ChDM interim materials (*p* < 0.05). Before and after all toothbrushing procedures, the CadPM specimens had smoother and ChPM specimens had rougher surfaces than the other interim materials. The two tested toothpastes had similar effects on the R_a_ of all interim materials. Non-perceivable color changes were seen only with the CadPM_Dw group.

## 1. Introduction

Interim crown materials are necessary to protect periodontal tissues, for prepared tooth, to maintain aesthetics, for chewing efficiency, and to ensure the positional stability of the tooth [1,2]. To fabricate interim crowns, either direct or indirect techniques, using poly methyl methacrylate (PMMA), polyvinyl ethyl methacrylate (PVEMA), poly ethyl methacrylate (PEMA), urethane methacrylate, bis-acryl compounds and bisphenol glycidyl methacrylate (bis-GMA) materials, can be used with different polymerization options [3].

Color stability is an important criterion for the selection of interim crown materials, especially in the anterior region [4,5]. Many factors such as the type, surface roughness, incomplete polymerization, and water sorption of the material, as well as the polishing technique used, oral hygiene, and the patient’s diet, can affect the magnitude of color alteration and thus patient satisfaction [3,4,5,6].

Surface finishing and polishing techniques are crucial for minimizing surface roughness, which may affect the definitive aesthetic appearance and the color of the restoration, plaque accumulation, biofilm adhesion, secondary caries, and periodontal health [1,2,7]. To achieve smooth surfaces, after finishing with abrasive stones and burs, water and fine pumice, polishing pastes or liquids, and silicone tips or surface sealant agents may be used [3,8,9,10].

Although the material surface becomes smoother after initial polishing, the surface roughness may increase in the oral environment, as the interim restoration may wear over time [11,12]. Toothbrush abrasion is amongst the causes for the potential wear on dental material surfaces [13]. In addition, while the use of toothpaste is crucial due to its therapeutic effect, it can increase the surface roughness of restorative materials and enamel [14]. Toothpaste abrasives and toothbrush bristles can disrupt the tooth and restorative material surface by forming superficial grooves. The rough surface can discolor and compromise the quality and success of the restoration. Therefore, it is particularly important to know the effects of toothbrushing and toothpaste on a restorative material’s surface [14,15]. Even though previous studies reported on the surface roughness and color stability of restorative materials, the effects of toothbrushing with various toothpastes on the surface roughness and color stability of different interim restorative materials have not been investigated to date.

This in vitro study aimed to evaluate the effects of toothbrush and toothpaste abrasion on the surface roughness and color stability of different interim crown materials. The first hypothesis of this study was that the material type would have a significant effect on the surface roughness and color stability after brushing with different toothpastes. The second hypothesis was that the effects of different toothpastes on the surface roughness and color stability of various materials would be different.

## 2. Materials and Methods

The surface roughness and color stability of 4 interim crown materials fabricated either chairside or during laboratory procedures were evaluated before and after 10,000 cycles of toothbrushing with distilled water and different toothpastes. The tested interim crown materials, surface sealant agents, and toothpastes are shown in Table 1.

Ten specimens per group were deemed appropriate from the calculations through ANOVA for the independent groups, with a 95% confidence (1-α), 80% test power (1-β), and f = 0.27 (for surface roughness) and f = 0.44 (for color) effect sizes. Forty disk-shaped (10 mm in diameter and 2 mm in thickness) specimens were prepared from each interim crown material. The mixing and polymerization processes of the Tab 2000 (ChPM), Imident (LaPM) and Protemp 4 (ChDM) specimens were carried out in a stainless-steel mold according to the manufacturers’ instructions. A disc-shaped (10 mm × 2 mm) wax pattern was scanned (Trios 3, 3Shape Inc., Copenhagen, Denmark) and the standard tessellation language (STL) data were transferred to a computer-aided design (CAD) software program (Ceramill Mind, Amanngirrbach AG, Koblach, Austria) for the fabrication of the CadPM specimens (Figure 1). The CadPM specimens were manufactured using a 5-axis milling machine (Ceramill Motion 2, Amanngirrbach AG) from Telio-CAD blocks.

All specimens were finished with a tungsten carbide rotary instrument (S194 190050, Horico, Berlin, Germany) and grounded in a sanding device (Phoenix Beta Buehler, Lake Bluff, IL, USA) at 100 rpm for 15 s with 400-grit silicon carbide abrasive paper (Atlas Waterproof Sheet, Saint-Gobain, Kocaeli, Turkey). Then, the specimens were ultrasonically cleaned in distilled water (Hygosonic, Dürr Dental AG, Bietigheim-Bissingen, Germany) for 10 min, rinsed, and dried in oil-free air. For a simplified, standardized polishing process and to obtain a high gloss, a surface coating agent (Optiglaze Color, OgC, Tokyo, Japan) was applied onto the surfaces of the specimens with a bristle brush in one direction and as a thin layer, without creating air bubbles, according to the manufacturers’ instructions. The polymerization of the coating agent layer was carried out for 90 s in a light-polymerizing unit (Laboligth Duo, GC Europe AG, Leuven, Belgium), without any tacky areas on the surfaces of the specimens.

To evaluate the effects of toothbrush abrasion and different kinds of toothpastes on the surface properties of the interim crown materials, specimens from each material group were divided into 4 subgroups by using a simple randomization technique (*n* = 10). While no toothbrushing was applied to the control group specimens (Cnt), the specimens of the 3 test groups were subjected to toothbrushing using a simulation device (DentArge TB-6.1, Analitik Medikal, Gaziantep, Turkey) for 10,000 cycles, which corresponds to 1 year of toothbrushing (Figure 2) [13]. The test groups consisted of a toothbrush (Banat Basic Medium, Banat Co., Istanbul, Turkey) and distilled water (Dw), a toothbrush and whitening toothpaste (Signal White Now, Unilever)-distilled water slurry (1:1) (WTp), and a toothbrush and activated charcoal toothpaste (Splat Blackwood, Splat Cosmetica, Moscow, Russia)-distilled water (1:1) slurry (ACTp). For each specimen, a new toothbrush and fresh slurries were used. The brushing action was performed at room temperature (25 °C) in a back-and-forth direction, and the standardization was achieved with a vertical force of 350 g, a stroke length of 10 mm, and cycle speed of 40 mm/s.

Surface roughness (R_a_) values of all specimens were measured using a contact profilometer (Perthometer M2; Mahr, Göttingen, Germany) with a 0.01 mm resolution, 0.8 mm interval (cut-off length), 5.5 mm transverse length, and 1 mm/s stylus speed. For each specimen, measurements were performed with the instrument’s diamond stylus (NHT-6) 3 times under constant pressure, and the mean R_a_ values were calculated and recorded in µm.

According to the CIE (Commission International de l’Eclairage) L*a*b* color parameters, initial color measurements for each specimen were taken using a digital spectrophotometer (VITA Easyshade, Vita Zahnfabrik, Bad Säckingen, Germany) 3 times, and the means were recorded as L_0_*, a_0_*, and b_0_*. After the initial measurements, the specimens were immersed in a stainless-steel container including staining solution, which was prepared by dissolving 7.5 g of coffee (Nescafe Classic, Nestle, Vevey, Switzerland) in 500 mL of boiled distilled water, according to the manufacturer’s suggested concentration. To simulate the intraoral conditions, the specimens were stored in this solution in a dark environment at 37 °C for 14 days, and the solution was changed every 24 h throughout the test [16,17]. Following the staining procedure, each specimen washed under running water, air-spray dried, and color measurements were repeated. The data were recorded as L_1_*, a_1_*, and b_1_*. Color change values were calculated using the CIEDE2000 (ΔE_00_) color difference formula [18,19,20]:ΔE_00_ = [(ΔL′/K_L_S_L_)^2^ + (ΔC′/K_C_S_C_)^2^ + (ΔH′/K_H_S_H_)^2^ + R_T_ (ΔC′/K_C_S_C_) (ΔH′/K_H_S_H_)]^1/2^

In the formula, while ΔL′, ΔC′, and ΔH′ represent the differences in hue, lightness, and chroma for a pair of specimens in the CIEDE2000, R_T_ is the rotation function that reports the interaction between the chroma and hue differences in the blue region. S_L_, S_C_, and S_H_ weighting functions adjust the total color difference for the variation in the location of the color difference pair in the L_0_, a_0_, and b_0_ coordinates. In the present study, the parametric factors K_L_, K_C_, and K_H_, which are the correction terms for the experimental conditions, were set to 1. The perceptible color difference and clinical acceptability threshold levels were set at 0.80 and 1.8, respectively [21].

The R_a_ and ΔE_00_ data were statistically analyzed. The Levene test of homogeneity was used to evaluate the distribution of the variables, and a normal distribution was observed. The R_a_ and ΔE_00_ results were analyzed separately with a two-way analysis of variance (ANOVA) for descriptive statistics and to evaluate the effects of the interim material type, the toothbrushing procedure, and their interactions. Multiple comparisons of the mean R_a_ and ΔE_00_ values were carried out with the Tukey’s HSD test. The significance was evaluated at *p* < 0.05 for all tests. All computations were performed with statistical software (IBM SPSS Statistics V20.0, IBM Corp., Armonk, NY, USA).

## 3. Results

According to the two-way ANOVA, the interim material type, toothbrushing procedure, and their interactions were significant for the R_a_ values (Table 2) (*p* < 0.05). The mean R_a_ values and standard deviations (SD) for the interim material and toothbrushing procedure combinations are shown in Table 3.

The mean R_a_ values for all groups (0.13 to 0.61 µm) were higher than the reported plaque accumulation threshold (0.20 µm), except for the CadPM specimens (Figure 3). While the highest R_a_ value was measured for the ChPM_WTp and ChPM_ACTp groups (0.61 ± 0.03 µm), the lowest R_a_ value was measured for the CadPM_Cnt group (0.13± 0.04 µm). For all interim material groups, even though all toothbrushing procedures caused an increase in the R_a_ values, statistically significant differences were observed only for WTp and ACTp applied to the ChPM, LaPM, and ChDM groups, and for Dw applied to the ChPM group, compared to the control group (*p* < 0.05). No statistically significant difference was observed between the R_a_ values of the WTp- and ACTp-applied groups, regardless of the interim crown material (*p* > 0.05). When the same brushing procedure was applied to groups and the results were compared, the R_a_ values of the ChPM groups were significantly higher, and the R_a_ values of the LaPM groups were statistically lower than all other interim material groups (*p* < 0.05). The R_a_ values of the Dw-, WTp-, and ACTp-applied ChDM groups were also higher than the LaPM and CadPM groups when the same toothbrushing procedure was applied (*p* < 0.05). However, the R_a_ values of the ChPM and ChDM test specimens brushed with distilled water were statistically significantly lower than those brushed with WTp and ACTp (*p* < 0.05).

The two-way ANOVA results showed that the interim material type, toothbrushing procedure, and their interactions were also significant in terms of the ΔE_00_ values (Table 2) (*p* < 0.05). Mean ΔE_00_ values and standard deviations (SD) for the interim material and toothbrushing procedure combinations are listed in Table 4.

The ΔE_00_ values of all test groups were above the visually perceptible range but still within clinically acceptable limits (0.80 < ΔE_00_ ˂ 1.80), except for the CadPM_Dw group (0.79 ± 0.19), which had a lower ΔE_00_ than the perceptibility threshold (ΔE_00_ ˂ 0.80) (Figure 4).

While the highest ΔE_00_ was observed for the ChDM_ACTp group (1.44 ± 0.16), the lowest was observed for the CadPM_Dw group (0.79 ± 0.19). When the control groups were compared, no significant difference was observed between the interim crown materials (*p* > 0.05). The ΔE_00_ values of WTp- or ACTp-applied ChPM and ChDM, and Dw-applied ChDM, were significantly higher than their control groups (*p* < 0.05). No statistically significant difference was found among the toothbrushing groups within the interim materials, except for the ChPM group (*p* > 0.05). Brushing with the WTp and ACTp toothpastes presented similar results in each interim material group (*p* > 0.05). Significant differences were observed between the Dw- and WTp-applied ChDM and CadPM specimens, and also between the ACTp-applied ChDM and LaPM, and all CadPM specimens (*p* < 0.05).

## 4. Discussion

The first hypothesis of the present study was accepted, because the material type had a significant effect on the surface roughness and color stability after brushing with different toothpastes. Because the effects of different toothpastes on the surface roughness and color stability of various materials were similar, the second hypothesis was rejected.

In the present study, a surface coating agent was applied to all interim crown materials to eliminate surface irregularities and to obtain smooth surfaces with a clinically straightforward and time-saving procedure [9,22]. Only the R_a_ values of the CadPM specimens were below the threshold value of the initial adhesion of dental plaque, which was reported by Bollen et al. [23] Previous studies showed that R_a_ values above 0.2 µm facilitate biofilm retention on the surfaces of definitive and interim restorations, contribute to the adsorption of color particles, and promote material wear [6,24,25]. The results of the present study revealed that polishing with a surface coating agent was insufficient for establishing an initial smooth surface on the ChPM, LaPM, and ChDM specimens, not resulting in values below the threshold value. Similar results were obtained in previous studies [1,3], where PMMA and bis-acryl specimens were polished with different techniques, and the R_a_ values of all specimens were below the clinical undetectability limit of 10 µm, which Kaplan et al. [26] identified.

According to the statistical analysis of the present study’s results, the interim material type was the most significant factor for the R_a_ values (F value = 564.716). Although it was prepared in accordance with the manufacturer’s instructions and by the same operator, the initially high R_a_ values observed in the self-polymerized chairside PMMA (ChPM), compared to the other test groups, can be attributed to the bubbles and irregularities formed depending on the manipulation technique (hand-mixed), which lacks adequate pressure and uniformity, as well as the composition and the particle size of the material [10]. However, other PMMA resins (LaPM and CadPM) had lower R_a_ values compared with those of the bis-acryl resin (ChDM). Similar findings were reported in previous studies [6,11,12], which attributed this outcome to the homogenous composition of acrylic resins and the heterogeneous design with respect to the particle size of the bis-acryl composites. The composition of the resin material, the presence of filler particles, their size distribution, as well as the chemistry of the resin can affect the polishability, and thus the surface roughness [12,27].

Toothbrush abrasion can cause increased surface roughness, plaque accumulation, a loss of gloss, and discoloration, which can reduce the aesthetic quality of a restoration [13,28,29]. Therefore, as the present study intended to achieve, it is important to evaluate the effects of toothbrushing on interim crowns that may be used for a long-time. For this purpose, in the current study, the toothbrush type, cycle rate, brushing force, and stroke length were standardized [13,29]. The results of the study showed that brushing with distilled water significantly increased the surface roughness of only the ChPM specimens.

An ideal toothpaste should clean with therapeutic functions and remove extrinsic staining without causing wear on the enamel and restorations. While toothpastes help to remove stains and discolorations by reducing the biofilm and calculus deposits during toothbrushing, they can also cause undesired abrasions on the teeth and restorative materials, depending on their main components and abrasive agents [30].

Whitening toothpastes, which have been reported to produce effective whitening compared to regular pastes [31], offer a practical and cost-effective whitening mechanism for both natural and restored teeth, working through the interaction between certain abrasive materials, such as calcium carbonate, hydrated silica, titanium oxide, or activated charcoal, combined with surfactants, enzymes, and peroxide compounds [18]. The effectiveness of these toothpastes varies depending on the shape, size, hardness, concentration, and distribution of the particles they contain. These pastes also affect the surface roughness of restorative materials, and consequently their aesthetic appearance and color, by increasing the surface porosities, promoting the loss of mass and water sorption [32,33].

In the present study, similar to a previous study [15], the use of WTp toothpaste containing hydrated silica with intermediate abrasive properties [34], and ACTp toothpaste containing activated charcoal that is highly porous, with the ability to adsorb chromophores and a high surface area [35], caused a similar increase in the surface roughness on ChPM and ChDM, and the differences compared to brushing with distilled water were significant for both resin materials (*p* < 0.05). When these toothpastes were used, the differences between the surface roughness values of all resin groups were statistically significant (*p* < 0.05), and the greatest effects were observed in the ChPM and ChDM interim resin groups, respectively.

In aesthetically critical areas, not only the color match of the provisional restoration, but also the continuity of its aesthetic appearance and color stability throughout the service period, are important. Therefore, perceptible color changes that may occur with long-term interim restorations jeopardize the acceptability of this restoration [5,36]. In the present study, according to the statistical analysis results, the brushing procedure was the most significant factor for the ΔE_00_ values (F value = 17,803), followed by the interim material type. Considering the 50:50% perceptibility threshold value of ΔE_00_:0.8, and the 50:50% acceptability threshold value of ΔE_00_:1.8, using the CIEDE 2000 system [21], the present study results revealed that only the ΔE_00_ values of the CadPM_Dw group were below the perceptibility threshold of 0.8, whereas the ΔE_00_ values of all other test groups were within clinically acceptable limits (˂1.8).

Exposure to food products, toothbrushes, toothpastes, and mouthwashes in daily life may increase the tendency for plaque adhesion and staining in the case of restorations with a greater surface roughness. Therefore, when the long-term use of interim restorations is required, a deterioration in periodontal health and aesthetic appearance may occur [11]. In the present study, all untreated PMMA-based and bis-acryl interim resin materials exhibited similar color changes in coffee. However, regardless of the slurry used, brushing caused an increase in the color change, along with the surface roughness, in the bis-acryl resin group. Although the bis-acryl resin tested in this study is available in cartridge auto-mixed systems, which reduce the amount of non-reactive monomer and water absorption, the resin showed considerable levels of color alteration compared to other specimens by acquiring stains from the immersion solution, as seen in previous studies [37,38]. This staining may have occurred because the resin is more polar than the PMMA, has a high diffusion coefficient, is hydrophilic, and has a heterogeneous composition that allows for water infiltration, leading to pigments infiltrating the interface between the fillers and the resin matrix [2,36,39].

The color stability of PMMA interim resin materials is affected by many factors, such as monomer–polymer conversion, monomer polarity, water sorption, pigment stability, the size distribution of PMMA particles, and the initiator system [38,40]. In the present study, compared with the other groups, the highest ΔE_00_ values were obtained with the chairside direct PMMA specimens (ChPM), with increasing Ra values observed after all toothbrushing procedures. In the PMMA group, the use of whitening dentifrices especially affected the color stability significantly. The presence of porosity, unreacted monomers, voids, and, hence, the water sorption capability may have changed the color of this PMMA [41]. In addition, it was observed that the LaPM specimens, which were prepared indirectly in the laboratory, provided better color stability after the application of all the brushing procedures. Differences in color stability between the ChPM and LaPM conventional resin groups may be due to the composition or particle size variables, which are also responsible for the differences in the surface roughness values [10].

In the current study, like the LaPM specimens, the CAD-CAM fabricated CadPM specimens provided better color stability and surface roughness values after brushing, as also observed in previous studies [38,42]. PMMA-based CAD-CAM blocks are pre-polymerized, have a low polymerization shrinkage, and high monomer–polymer conversion, and they are more advanced in terms of their mechanical and surface properties compared to manually produced resins [38,43]. Therefore, CAD-CAM interim materials, thanks to their improved durability and enhanced color stability, may be preferred, especially in aesthetic terms, when long-term use is required [38].

This in vitro study has some limitations. The flatness of the specimens and the absence of anatomical grooves and pits do not completely reflect real clinical polishing or plaque control challenges. In addition, some variables in the oral environment, such as salivary proteins and enzymes, temperature changes, dietary habits, smoking, and functional or parafunctional loads, should be considered in future studies.

## 5. Conclusions

Within the limitations of this study, it was concluded that, except for the CadPM specimens, all interim resin materials had surface roughness values higher than the plaque accumulation threshold (0.20 µm) before and after toothbrushing, regardless of the toothpaste used. Moreover, toothbrushing with toothpastes containing activated charcoal and hydrated silica increased the surface roughness, except in the CadPM specimens.

The ΔE_00_ values of all resin specimens immersed in coffee (except for the CadPM_Dw specimens) exceeded the 50:50% perceptibility threshold value but were within clinically acceptable limits. Toothbrushing with both toothpastes increased the ΔE_00_ values of the ChPM and ChDM interim resin materials similarly. Moreover, it was concluded that the LaPM and CadPM specimens provided better color stability after brushing with distilled water or dentifrices.

## Figures and Tables

**Figure 1 materials-15-05831-f001:**
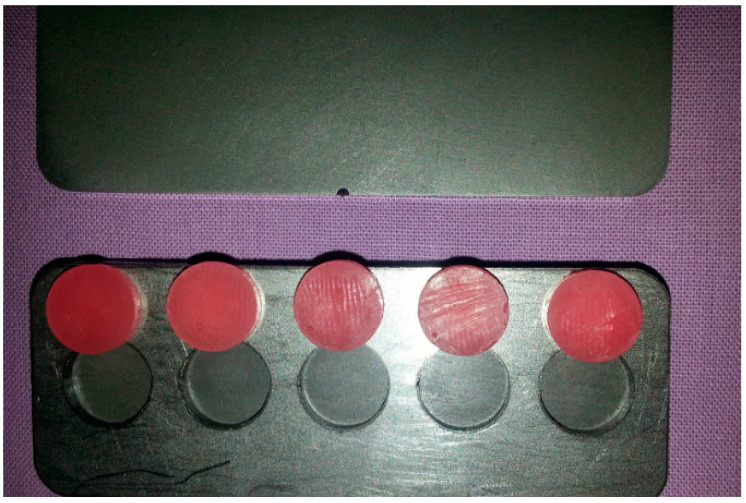
Stainless-steel mold used in the study and disc-shaped wax patterns for CadPM specimens.

**Figure 2 materials-15-05831-f002:**
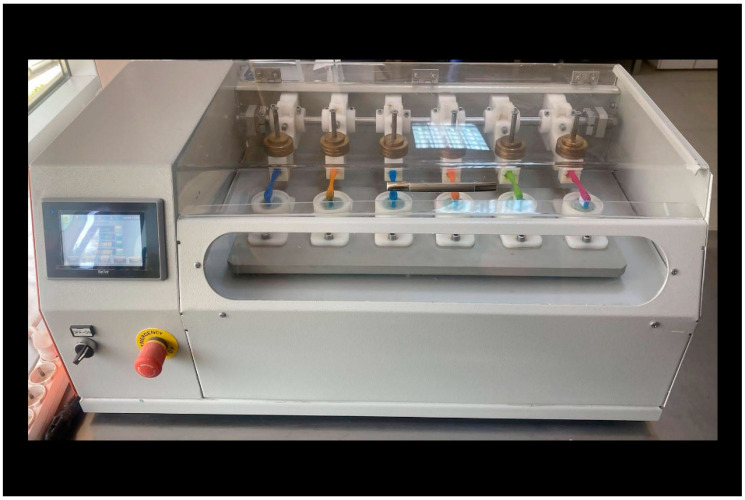
Toothbrushing simulation device used in the study.

**Figure 3 materials-15-05831-f003:**
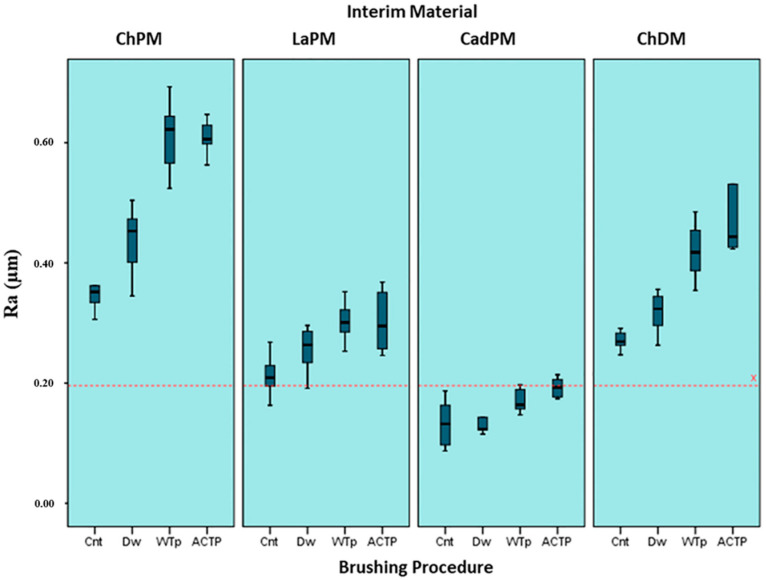
Mean R_a_ (±SD) values of the test groups. The plaque accumulation threshold (R_a_ = 0.2 mm) is indicated as line-x.

**Figure 4 materials-15-05831-f004:**
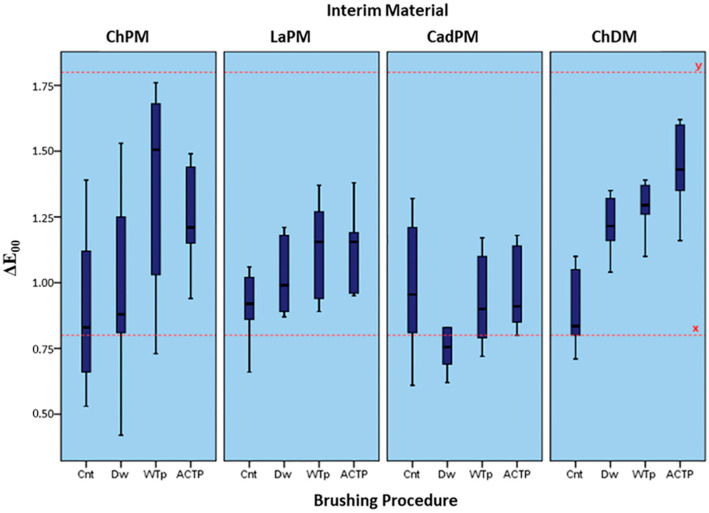
Mean (±SD) ΔE_00_ values of the test groups. The perceptibility threshold of the color differences (ΔE_00_ = 0.8) is indicated as line-x, and the acceptability threshold (ΔE_00_ = 1.8) as line-y.

**Table 1 materials-15-05831-t001:** Materials used in the study.

Code	Material	Type	Components	Manufacturer
ChPM	Tab 2000	Chairside direct polymethyl methacrylate resin	Methyl methacrylate, *n*-butylmethacrylate	Kerr Corp. (Scafati, Italy)
LaPM	Imident	Laboratory indirect polymethyl methacrylate resin	Polymethyl methacrylate powder (cadmium free), methyl methacrylate monomer	Imicryl (Konya, Turkey)
CadPM	Telio CAD	CAD CAM cross-linked polymethyl methacrylate	Polymethyl methacrylate	Ivoclar Vivadent AG (Schaan, Liechtenstein)
ChDM	Protemp 4	Chairside direct bis-acryl composite resin	Ethanol,2,2′-[(1-methylethylidene)bis(4,1-phenyleneoxy)]bis-, diacetate, benzyl-phenyl-barbituric acid, silane treated silica, tert-butyl peroxy-3,5,5-trimethylhexanoate	3M ESPE (Seefeld, Germany)
OgC	Optiglaze Color	Surface coating agent	Methyl methacrylate, polymethyl methacrylate, silica filler, photo inhibitor	GC Corp. (Tokyo, Japan)
WTp	Signal White Now	Whitening toothpaste	Hydrogenated starch hydrolysate, aqua, hydrated silica, sodium Lauryl sulfate, PEG 32, aroma, cellulose gum, sodium fluoride, sodium saccharin, PVM/MA copolymer, mica, trisodium phosphate, glycerin, sodium lauryl sulfate, lecithin, caprylyl Glycol, limonene, CI 74160, CI 77891	Unilever (Rueil-Malmaison, France)
ACTp	Splat Blackwood	Activated charcoal toothpaste	Aqua, hydrated silica, hydrogenated starch hydrolysate, glycerin, maltooligosyl glucoside, sodium lauroyl sarcosinate, cellulose gum, aroma, charcoal powder, capryloyl/caproly methyl glucamide, lauroly/myristoyl methyl glucamide, sodium sorbate, menthol, o-cymen-5-ol, Juniperus communis sprout extract, limonene	Splat-Cosmetica (Moscow, Russia)

**Table 2 materials-15-05831-t002:** Two-way ANOVA results for R_a_ and ΔE_00_ values.

Source		SS	df	MS	F	*p **
**R_a_**	Interim material (A)	2.582	3	0.861	564.716	<0.001 *
	Brushing procedure (B)	0.611	3	0.204	133.686	<0.001 *
	AxB	0.203	9	0.023	14.761	<0.001 *
	Error	0.219	144	0.002		
	Total	20.575	160			
**ΔE_00_**	A	1.802	3	0.601	14.839	<0.001 *
	B	2.162	3	0.721	17.803	<0.001 *
	AxB	1.627	9	0.181	4.467	<0.001 *
	Error	5.828	144	0.040		
	Total	195.760	160			

SS, sum of squares; df, degrees of freedom; MS, mean square; F, F value (variation between the sample means/variation within the samples). * *p* < 0.05 indicates statistical significance.

**Table 3 materials-15-05831-t003:** Mean R_a_ values (µm) and standard deviations (SD) of test groups with Tukey’s HSD multiple comparisons.

Interim Material	ChPM	LaPM	CadPM	ChDM
**Brushing Procedure**	Mean ± SD *	Mean ± SD *	Mean ± SD *	Mean ± SD *
Cnt	0.36 ± 0.05 ^Ca^	0.21 ± 0.03 ^Ba^	0.13± 0.04 ^Aa^	0.27 ± 0.03 ^Ba^
Dw	0.44 ± 0.05 ^Db^	0.26 ± 0.03 ^Bab^	0.15 ± 0.05 ^Aa^	0.32 ± 0.03 ^Ca^
WTp	0.61 ± 0.03 ^Dc^	0.31 ± 0.03 ^Bb^	0.17 ± 0.02 ^Aa^	0.42 ± 0.05 ^Cb^
ACTp	0.61 ± 0.03 ^Dc^	0.30 ± 0.04 ^Bb^	0.19 ± 0.01 ^Aa^	0.47 ± 0.05 ^Cb^

* Tukey’s HSD test results are shown as letters, and there is no statistically significant difference between values indicated by the same letter (*p* > 0.05). While lowercase letters show differences between toothbrushing groups for the same interim crown material, uppercase letters indicate differences when the same brushing procedure was applied to the interim crown materials.

**Table 4 materials-15-05831-t004:** Mean ΔE_00_ values and standard deviations (SD) of test groups with Tukey’s HSD multiple comparisons.

Interim Material	ChPM	LaPM	CadPM	ChDM
**Brushing Procedure**	Mean ± SD *	Mean ± SD *	Mean ± SD *	Mean ± SD *
Cnt	0.88 ± 0.29 ^Aa^	0.92 ± 0.12 ^Aa^	1.00 ± 0.23 ^Aa^	0.88 ± 0.14 ^Aa^
Dw	0.98 ± 0.32 ^ABab^	1.03 ± 0.15 ^Aba^	0.79 ± 0.19 ^Aa^	1.22 ± 0.10 ^Bb^
WTp	1.36 ± 0.37 ^Bc^	1.12 ± 0.18 ^Aba^	0.93 ± 0.16 ^Aa^	1.31 ± 0.13 ^Bb^
ACTp	1.24 ± 0.17 ^Abbc^	1.13 ± 0.14 ^Aa^	0.96 ± 0.14 ^Aa^	1.44 ± 0.16 ^Bb^

* Tukey’s HSD test results are shown as letters, and there is no statistically significant difference between values indicated by the same letter (*p* > 0.05). Lowercase letters indicate differences between toothbrushing groups for same interim crown material, and uppercase letters indicate differences among interim crown material groups when the same brushing procedure was applied.

## Data Availability

Not applicable.

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
