# Peer review of "Influences of Toothbrushing and Different Toothpastes on the Surface Roughness and Color Stability of Interim Prosthodontic Materials"

_materials, 2022, doi:10.3390/ma15175831_

Round 1

Reviewer 1 Report

Manuscript ID: materials-1842859

This manuscript, Influence of toothbrushing and different toothpastes on the surface roughness and color stability of interim prosthodontic materials, is an interesting paper focusing on the effect of toothbrushing with different kinds of toothpaste and color stability of different interim crown materials. The paper is well-thought, especially due to standardized tests (cycle rate, brushing force and stroke length, which allow for the comparison of various pastes as well as materials. Moreover, the manuscript has a very well-prepared discussion with valuable scientific conclusions. The weaknesses of the article include only the lack of visualization (illustrative photo of the tested materials, toothbrushing test stand and/or the effects of color change after stability testing). Of the important things that need improvement, describe the statistical symbols under each table (Table 2, Table 3 and Table 4). Additionally, I would like to ask a question for a scientific discussion - why distilled water was used in the toothbrush abrasion tests and a real conditions have not been simulated (e.g. artificial salive solution + tap water as well as temperature).

Reviewer 2 Report

Dear authors,

congratulations for the chosen subject; provisional prosthetic restorations have an important role not only from an aesthetic but also a functional point of view.

To facilitate the understanding of some details, please specify the following:

-which type of toothpaste is recommended to preserve the qualities of temporary restorations?

- does the processing and finishing of temporary restorations have to follow a specific protocol?

- which of the analyzed characteristics do you consider more important - roughness or color, in the case of provisional restorations?

Kind regards!

Reviewer 3 Report

- Please identify the abbreviation used in the abstract (WTp, ACTp) for the reader. 

- Please provide components of Imident e.g. type of monomers, initiator, and activator 

- Please ensure that the equation of color difference (DeltaE) is inserted as an equation, not a figure. 

- DelE00; the "00" should be subscripted similar according to the equation 

- I think the Ra values can be presented as numerical values with only two decimals, similar to the result of color stability. 

- Figures 1 and 2 are box-plot charts. Please confirm whether they are mean (SD) or median (min-max) with quartile ranges.  The box-plot chart usually presents the median (min-max) for non-normally distributed data. 

- please check all typos throughout again. 

- I would suggest revising the conclusion into statements, not numbers or bullet points. Additionally, the statistical results should not be included in this part.  

Reviewer 4 Report

The paper can be considered once the author include some pictures to illustrate the the experiment setup.
